# Current Paradigms of Tolerogenic Dendritic Cells and Clinical Implications for Systemic Lupus Erythematosus

**DOI:** 10.3390/cells8101291

**Published:** 2019-10-21

**Authors:** Patcharee Ritprajak, Chamraj Kaewraemruaen, Nattiya Hirankarn

**Affiliations:** 1Research Unit in Integrative Immuno-Microbial Biochemistry and Bioresponsive Nanomaterials, Faculty of Dentistry, Chulalongkorn University, Bangkok 10330, Thailand; Patcharee.R@chula.ac.th; 2Department of Microbiology, Faculty of Dentistry, Chulalongkorn University, Bangkok 10330, Thailand; 3Center of Excellence in Immunology and Immune-Mediated Diseases, Chulalongkorn University, Bangkok 10330, Thailand; kchamraj@gmail.com; 4Immunology Unit, Department of Microbiology, Faculty of Medicine, Chulalongkorn University, Bangkok 10330, Thailand

**Keywords:** tolerogenic dendritic cells, phenotype, function, metabolic control, clinical implication, systemic lupus erythematosus

## Abstract

Tolerogenic dendritic cells (tolDCs) are central players in the initiation and maintenance of immune tolerance and subsequent prevention of autoimmunity. Recent advances in treatment of autoimmune diseases including systemic lupus erythematosus (SLE) have focused on inducing specific tolerance to avoid long-term use of immunosuppressive drugs. Therefore, DC-targeted therapies to either suppress DC immunogenicity or to promote DC tolerogenicity are of high interest. This review describes details of the typical characteristics of *in vivo* and *ex vivo* tolDC, which will help to select a protocol that can generate tolDC with high functional quality for clinical treatment of autoimmune disease in individual patients. In addition, we discuss the recent studies uncovering metabolic pathways and their interrelation intertwined with DC tolerogenicity. This review also highlights the clinical implications of tolDC-based therapy for SLE treatment, examines the current clinical therapeutics in patients with SLE, which can generate tolDC in vivo, and further discusses on possibility and limitation on each strategy. This synthesis provides new perspectives on development of novel therapeutic approaches for SLE and other autoimmune diseases.

## 1. Introduction

Dendritic cells (DCs) are antigen-presenting cells that act as immune sentinels, and play a central role in activation of immune responses and in mediation of immune homeostasis and immune tolerance [1,2]. The functional versatility and plasticity of DCs are influenced by multiple stimuli and microenvironments [3,4,5,6]. Immunogenic DCs may be primarily induced by pathogens inflammatory signal molecules, and they typically promote effector function of adaptive immune cells [1,7]. On the other hand, tolerogenic DCs (tolDCs) are induced by either anti-inflammatory signals or signals interfering with immunogenic DC function, and they play an important role in induction of immune tolerance, resolution of ongoing immune responses and prevention of autoimmunity by inhibiting effector and autoreactive T cells, and by triggering regulatory T cell (Treg) development [7,8,9]. The common features of tolDCs are low maturation, reduced pro-inflammatory cytokine production, increased inhibitory molecule expression, enhanced anti-inflammatory cytokine production, and increased ability to induce Treg as well as immune tolerance [6,7,8].

Over the past several decades, tolDCs have become promising cell-based therapies for treatment of autoimmune diseases including systemic lupus erythematosus (SLE) [10,11,12,13]. Various pharmacological and biological agents have been employed to generate tolDCs *ex vivo* for DC-based immunotherapy [5,6,14]. Here, we will also propose that targeting DCs *in vivo* is an alternative strategy to skew DCs toward tolerogenic phenotypes. The characteristics and properties of tolDCs can vary depending on the tolDC-inducing protocol [14,15,16]. Furthermore, the phenotypic and functional features of tolDCs required for effective therapy may differ based on the pathogenesis of distinct autoimmune diseases [14,17]. In this review, we discuss our current understanding of tolDCs and highlight some clinical implications for SLE treatment.

## 2. DC Subsets in Immune Tolerance

DCs are heterogeneous in phenotype and function, and specialized subsets of DCs can orchestrate many different types of T cell responses. DCs are principally classified into two major populations: conventional DCs (cDCs) and non-conventional DCs, including plasmacytoid DCs (pDCs) and monocyte-derived (moDCs). DCs originate from bone marrow hematopoietic stem cells (HSCs) that develop to macrophage and DC precursors (MDPs), and MDPs further give rise to common DC precursors (CDPs) and monocytes [18]. CDPs are differentiated to pDCs in bone marrows, and pre-DCs which migrate to lymphoid and non-lymphoid tissues and differentiated to lymphoid resident cDCs and migratory cDCs, respectively [19]. cDCs are distinguished by expression of the transcription factor zinc finger and BTB domain containing 46 (Zbtb46) [20,21], and are further categorized into type 1 cDCs (cDC1s) and type 2 cDCs (cDC2s). Lineage commitment in cDCs requires distinct transcription factors: basic leucine zipper transcriptional factor ATF-like 3 (BATF3) and interferon regulatory factor (IRF) 8 for cDC1s [22,23], and IRF4 for cDC2s [23,24]. pDCs uniquely express the transcription factor, E-protein transcription factor 4 (TCF4 or E2-2), which is a specific regulatory for pDC development [25]. Mouse and human DC ontogeny and development have been studied in detail, and the mechanisms involved in immune tolerance vary among DC subsets (Table 1). The phenotypes and functions of distinct subtypes of human DCs are less clear owing to the limitations of human *in vivo* studies.

Murine cDCs express high levels of CD11c and various levels of MHC class II [44]. Each subset can be identified using additional markers. Based on origin and location as mentioned above, murine cDCs are classified into lymphoid tissue-resident cDCs and non-lymphoid or migratory cDCs. Lymphoid tissue-resident cDC1s (in spleen and LNs) uniquely expresses CD8α while migratory cDC1s are defined by expression of CD103 [45]. Resident and migratory cDC1s share the phenotypic markers Clec9a (C-type lectin domain family member 9) (DNGR1, Dendritic cell natural killer dectin group receptor 1), XCR1 and CD24 [1,45]. The cDC1 subset can potently activate cytotoxic T cells via their cross-presentation capability [46,47]. However, these cells also have immunoregulatory functions and mediate immune tolerance to overcome autoimmune reactivity [2,26]. IFN-γ (Interferon gamma) stimulated CD8α^+^ cDC1s highly expressed indoleamine 2,3-dioxygenase (IDO) and these DCs suppressed T cell proliferation and induced T cell apoptosis [26,27]. A special subset of cDC1, BTLA (B and T lymphocyte attenuator)^+^CD205^+^CD8α^+^ DCs specifically mediated induction of antigen-specific FoxP3^+^ Tregs through B and T lymphocyte attenuator (BTLA) and transforming growth factor (TGF)-β [28]. Although, the mechanism of Treg induction by CD103^+^ cDC1s remains unclear, the well-known CD103^+^CD11b^-^ DCs in intestinal tissues are also a specialized cDC1 subset that expresses IDO and governs mucosal Treg differentiation via a TGF-β- and retinoic acid (RA)-dependent mechanism [32,33]. Human peripheral blood CD141^+^ DCs (or BDCA3^+^ DCs) are the counterparts of murine CD8α^+^ and CD103^+^ cCD1s [48]. Although there is no direct evidence showing tolerance induction by circulating CD141^+^ cDC1s, several lines of evidence link them with immunoregulatory functions. For example, CD141^+^CD163^+^ DCs in skin constitutively produced interleukin (IL)-10 and induced potent Tregs that inhibited skin inflammation [37]. Similarly, 1,25-dihydroxyvitamin D3 (vitD)-treated human blood CD141^+^ DCs manifested immunoregulatory properties [37]. Another distinct subset of human peripheral blood IL-10-producing DCs (DC-10s) are characterized by the markers CD14, CD16, CD11c, CD11b, HLA-DR, and CD83, and potently induced IL-10-producing Tregs dependent on ILT4/HLA-G signaling [38]. Recently, human DC-10s have been found to highly express CD141 and CD163, and DC-10s generated *in vitro* by treatment of MoDCs with IL-10 exhibited a similar tolerogenic signature to *in vivo*-derived DC-10s [39].

Murine lymphoid tissue-resident cDC2s have a CD8α^−^CD11b^+^DCIR2^+^SIRPα^+^ phenotype, and promote polarization of T helper (Th) 2 responses [45,49,50]. Migratory cDC2s are identified as CD103^−^CD11b^+^CCR2^+^SIRPα^+^ cells and also express additional markers such as CX3CR1 and CD301b [51,52]. This subset drives Th2 and Th17 responses [50,53,54]. Several lines of evidence pointed toward a role of cDC2s in immune tolerance. Resident CD11b^+^DCIR2^+^ cDC2s promoted sustained expansion of FoxP3^+^ Tregs in autoimmune diabetic mice [29]. One mechanism through which this cDC2 subset mediated immune tolerance was via inhibition of dendritic cell inhibitory receptor (DCIR)2 signaling [30]. Skin-derived CD103^−^CD11b^+^ cDC2s induced FoxP3^+^ Tregs by exploiting aldehyde dehydrogenase (ALDH) activity and secreting RA [34]. Furthermore, migratory cDC2s that expressed programmed death ligand-2 (PD-L2 or B7-H2 or CD273) were able to generate TGF-β-producing FoxP3^−^ LAP^+^ Tregs [35]. Human circulating cDC2s are CD1c^+^ (BDCA1^+^) and appear to be more closely related to mouse CD11b^+^ cDC2s [23,55]. The CD1c^+^ cDC2 subset also resides in skin and mucosal tissues, and is responsible for Th17 induction [56,57]. Similar to murine cDC2s, the immunoregulatory functions of CD1c^+^ cDC2s in response to infection and inflammation have been elucidated. Toll-like receptor (TLR)4-expressing cDC2s highly produced IDO and IL-10 in response to *Escherichia coli*, and these DCs could inhibited T cell proliferation dependent on IL-10 under TLR4 signal transduction [40]. Human pulmonary cDC2s mediated IL-10 producing Treg, which suppressed chronic obstructive pulmonary disease (COPD), via IL-10, IL-27 and inducible T cell costimulator ligand (ICOSL) [41]. 

pDCs are specialized DCs that predominantly produce type I interferon (IFN) upon pathogen encounter. Murine pDCs are characterized by low expression of CD11c and MHC class II and expression of B220, Siglec-H, and BM stromal cell antigen 2 (BST2; CD137; plasmacytoid dendritic cell antigen-1 or PDCA-1) [45]. The conventional markers of human pDCs are CD45RA, CD123 (I-3R), CD303 (CLEC4C; BDCA2), and CD304 (neuropilin; BDCA4) [58]. Mouse and human pDCs are implicated in peripheral tolerance, especially in the steady state, resulting in prevention of inflammation and autoimmunity [36,59]. Murine and human IDO^+^ pDCs in aorta have been found to be tolerogenic pDC subsets that protect against atherosclerosis via Treg induction [36]. In addition, engagement of TLR9 on pDCs could induce IDO expression and leaded to generation of Tregs [42,43].

Other distinct DC subsets, such as CD19^+^ splenic DCs and perforin-expressing DCs, are also capable of limiting inflammation and increasing resistance to autoimmune diseases [60,61]. Epidermal Langerhans cells (LCs) are sometimes classified as a DC subset. However, transcriptome analyses revealed that LCs are unique antigen-presenting cells in between DCs and macrophages [62,63]. The immunosuppressive effects of LCs have been extensively studied in several immune-mediated inflammatory diseases. LCs maintained immune tolerance in skin by activating skin-resident Tregs [64], and regulated local tissue tolerance in SLE by preventing autoantibody accumulation in the skin [65].

Immunogenic DCs can be re-directed to tolDCs depending on environmental factors such as tumor microenvironment, infectious agents, and some specific stimuli (e.g. pharmacological agents and biological molecules) [3,4,5,6]. The distinct anatomical site, especially the high immunosuppressive environment in mucosal tissues, also contributes to tolerogenic features of DC subsets [66]. Nevertheless, a certain condition involved in a fate of some specialize tolerogenic DC subsets (e.g., BTLA^+^CD205^+^CD8α^+^ DCs, PD-L2^+^ migratory cDC2s, etc.) yet has to be further elucidated. Several *in vivo* tolDCs express cell surface inhibitory molecules, such as BTLA and DCIR, which can be used to identify these DC subsets [28,30]. Of note, some surface molecules (e.g. TLR4) generally involved in DC inflammatory responses are able to transduce tolerogenic signals under specific intrinsic factors (e.g. IRF4 in cDC2) which expressed in certain DC subsets [40]. There is no clear evidence supporting the conversion of tolerogenic DCs to immunogenic DCs, however, loss of DC tolerogenicity have been shown to relate to genetic disorders and genetic variants in DC regulatory molecules, which partly contribute to the autoimmune disease development and pathogenesis [66,67].

## 3. Phenotypic and Functional Signatures of *Ex Vivo* Generated tolDCs

Various pharmacological agents and biological molecules can be used to generate tolDCs *in vitro* and *in vivo*. Their effects and mechanisms in modulating the phenotypic and functional features of tolDCs have been extensively discussed [5,6,8]. DC tolerogenicity of is determined by DC phenotype, function, and maturation status, all of which affect the ability of DCs to induce immune tolerance by triggering anti-inflammatory responses, inhibiting cell-mediated immune responses, and promoting Treg generation and expansion. Recent studies have extensively analyzed the transcriptional profiles of *in vitro*-generated tolDCs with the goal of identifying potential biomarkers for quality control and successful generation of tolDCs in clinical trials [15,68,69,70]. However, the most important concern in tolDC induction is to obtain sufficient functional quality for clinical therapy of distinct autoimmune diseases in individual patients. In addition, DC-associated molecules directly involved in modulating immune activation and tolerance should be down- and up-regulated, respectively, under tolerogenic conditions (Figure 1).

The key success of exploiting tolDCs in autoimmune disease therapy is the functional quality of tolDCs which is determined by the maturation stage, cytokine profiles, expression of inhibitory molecules, expression of chemokine and chemokine receptors. Furthermore, metabolic profiles may be used as certain tolDC biomarkers. PGE_2_, prostaglandin E2; IL-1RA, interleukin-1 receptor antagonist; C1q, complement 1q; THBS1, thrombospondin-1; PD-Ls, programmed death-ligands; ICOSL, inducible T-cell co-stimulator ligand; BTLA, B and T lymphocyte attenuator; CTLA-4, cytotoxic T-lymphocyte-associated protein 4; PD-1, programmed death-1; FASL, Fas ligand; ILTs, inhibitory receptors Ig-like transcripts; IDO, indoleamine 2,3-dioxygenase; HO-1, heme oxygenase-1; FAO, fatty acid oxidation; OXPHOS, oxidative phosphorylation; ALDH, aldehyde dehydrogenase.

T cell activation and function can be positively or negatively regulated by co-signaling molecules. TolDCs typically have downregulated expression of T cell-activating molecules such as CD40, CD80, and CD86, resulting in inability to prime and activate T cells [71]. Additionally, the enriched expression of immune checkpoint molecules, such as programmed death-ligands (PD-Ls), BTLA and inducible T-cell co-stimulator ligand (ICOSL), on tolDCs enables these cells to inhibit effector T cells and trigger Treg induction [28,72,73]. Cytotoxic T-lymphocyte-associated protein 4 (CTLA-4) is a negative regulator that prevents aberrant T cell responses against self-antigens. CTLA-4 is expressed on activated T cells and is also inducible on human MoDCs. Engagement of CTLA-4 on DCs led to downregulation of IL-8 and IL-12 production but upregulation of IL-10 production [74]. Another important T cell inhibitory receptor, programmed cell death protein-1 (PD-1), is also expressed on mature murine splenic DCs, and plays a crucial role in regulating DC function and T cell suppression [75]. Induction of PD-1 expression on DCs *in vitro* is selectively mediated by engagement of TLR ligands [76].

High expression of inhibitory receptors Ig-like transcripts (ILTs), such as ILT2, ILT3 and ILT4, has been detected on DCs differentiated *in vitro* under several tolerogenic conditions [77]. Activation of ILTs promoted tolerogenicity of DCs and subsequent T-cell suppression [77]. The expression of Fas ligands (CD95L) on DCs through genetic modification successfully inhibited T cell responses. However, investigations of FasL expression have been restricted to *in vivo*-derived DC subsets [78], and there is still no clear evidence to directly demonstrate expression of FasL on *in vitro*-generated tolDCs. An immunoregulatory receptor, FcγRIIB, was implicated in maintenance of immune tolerance as mice lacking FcγRIIB spontaneously developed an immune-mediated pathology similar to SLE [79]. FcγRIIB on DCs contributes to regulation of B and T cell responses [80,81]. FcγRIIB expression on DCs is often upregulated in response to pathogens or inflammatory stimuli. Moreover, transcriptome studies of human tolDCs induced by IL-10, TGF-β and dexamethasone demonstrated increased expression of the *FCGR2B* gene [82,83]. Another interesting molecule expressed on tolDCs and inducible by vitD or dexamethasone is CD300LF (CD300F, IREM-1 or LMIR3) [83,84]. CD300LF contains both an immunoreceptor tyrosine-based activating motif and an immunoreceptor tyrosine-based inhibitory motif in its long cytoplasmic domain, which can positively and negatively regulate immune cell function [85,86]. CD300L is expressed on various immune cells including B cells, macrophages, neutrophils and DCs, and depletion of CD300LF leads to autoimmune disease progression [86,87]. The functions of monocytes and neutrophils could be inhibited via CD300LF activation [88,89], and macrophages lacking CD300LF were unable to phagocytose apoptotic cells [87]. However, the role of CD300LF on DCs remains to be elucidated.

Expression of chemokine receptors is essential for tolDC migration and localization. CCR7 and CXCR4 are primarily responsible for DC migration to LNs and are upregulated upon DC maturation [84]. Lymphoid tissue homing receptors are required for tolDC migration to enable interactions with T cells residing in lymphoid organs. However, CCR7 and CXCR4 are often downregulated by DCs under tolerogenic conditions. Therefore, addition of inflammatory stimuli, such as lipopolysaccharide (LPS) and monophosphoryl lipid A, is required for induction of these chemokine receptors on tolDCs [90,91]. Sequential treatment with vitD and dexamethasone enhanced the expression of CCR5, which enabled tolDCs to migrate to peripheral tissues and across the blood-brain barrier [92,93]. Chemokine ligands also play a significant role in immune cell recruitment and activation. The Treg chemotactic factors, CCL18 and CCL22, were predominantly upregulated in tolDCs induced by dexamethasone and vitD, respectively [39]. Intriguingly, CCL18 plays an additional role in development of tolDCs with a semi-mature phenotype [94], and a recent study demonstrated that CCL22 promoted sustained DC-Treg contacts in human LNs, enabling Treg priming by DCs [95]. TolDC generation in the presence of IL-10 induced upregulation of CXCL1 [82,96], and the resulting neuroprotective effect contributed to improvement of autoimmune encephalomyelitis [97,98].

Other phenotypic features of tolDCs that mediate immune tolerance are reduced production of pro-inflammatory cytokines such as IL-12 and augmented production of anti-inflammatory cytokines such as IL-10 and TGF-β [99]. Transcriptome analysis has unveiled some immunoregulatory molecules upregulated by *in vitro*-generated tolDCs, including IL-1 receptor antagonist (IL-1RA), complement component 1q (C1q), thrombospondin-1 (THBS1 or TSP1), and mucin-like protein 1 (MUCL1) [7,16,70,84,93,96]. IL-1RA competitively blocks binding of IL-1 proteins to their receptors, and can impede DC maturation and priming of T cell responses [100]. C1q promoted DC phagocytosis of apoptotic cells and enhanced differentiation of IL-10-secreting CD4^+^ T cells [101,102]. DC-derived THBS1 promoted high production of prostaglandin E2, IL-10 and TGF-β by DCs, and the ligation of THBS1 with its receptor CD47 promoted Treg development in the periphery [103,104]. *Mucl1*-deficient DCs displayed a hyper-responsive phenotype upon LPS stimulation; therefore, MUCL1 may act as a negative regulator of DCs [105].

## 4. Metabolic Control of DC Tolerogenicity

Fine-tuning of cellular metabolism has become an important paradigm for modulating DC function. Changes in cellular metabolism, including amino acid catabolism, glycolysis, oxidative phosphorylation (OXPHOS) and fatty acid oxidation (FAO), are intimately associated with DC function. Better understanding of the roles of metabolic processes in DC tolerogenicity will accelerate the development of novel strategies to manipulate DCs for clinical therapy.

### 4.1. Amino Acid Metabolism

IDO and tryptophan catabolism have emerged as central hubs of the metabolic adaptations of tolerogenic DCs [106]. Tryptophan is catabolized into kynurenine by the intracellular enzyme, IDO. Elevated IDO activity contributed to tolerogenic properties of DCs which promoted Treg induction and immune suppression, and consequently leading to amelioration of autoimmune diseases [107,108,109]. Differential IDO expression has been observed among various DC subsets. Murine splenic cDC2s expressed IDO at very low levels, while CD8α^+^ cDC1s expressed higher levels. In addition, IFN-γ potentially activates the tolerogenic programs of cDC1s in an IDO-dependent fashion, regulated by interferon-stimulated response elements (ISREs) and gamma interferon activation sites in concert with the transcription factor IRF8 [26,27,110,111]. cDC2s can also become tolerogenic upon IFN-γ stimulation, but through mechanisms downstream of the IDO, a kynurenine pathway, and independently of tryptophan deprivation [31]. A recent proteomic study of human blood DC subsets detected unique expression of IDO in CD141^+^ cDC1s, but not in CD1^+^ cDC2s [112]. Human blood cDC1s and murine CD8α^+^/CD103^+^ cDC1s share several phenotypic and functional features, including high expression of IRF8 [48,113]. Expression of IDO was also detected in human and murine intestinal CD103^+^ DCs that play a pivotal role in induction of Tregs and mucosal tolerance [33,114]. Similarly to cDCs, murine and human pDCs employ IDO to induce Treg differentation and to prevent autoimmunity [43,115]. Upregulation of IDO in pDCs depends on type I IFN signaling pathways and ISREs [110,115,116].

An intracellular enzyme, heme oxygenase-1 (HO-1), degrades heme groups and its metabolites mediate the immunosuppressive properties of tolDCs. Immature DCs express HO-1 and this expression rapidly decreases upon DC maturation [117]. HO-1-expressing DCs promoted induction of antigen-specific Foxp3^+^ Tregs and decreased tissue inflammation [118]. Induction of high HO-1 expression in immature DCs by HO-1 inducers imbued these DCs with more potent tolerogenic properties, resulting in attenuation of autoimmunity and severe inflammation [118,119]. Murine splenic cDC1s expressed high levels of HO-1 that were correlated with high levels of IDO [120]. Furthermore, HO-1-mediated IDO expression contributed to the phenotypes and functions of tolDCs [121].

Arginase is an immunoregulatory enzyme that hydrolyzes L-arginine to L-ornithine and urea [122]. Two arginase isoforms, arginase 1 and arginase 2, have been identified in mammalian cells, and each isoform is encoded by different genes. These two isoforms share structural homology and functional properties [123]. Arginase 1 was markedly upregulated in DCs following IL-4 and TGF-β stimulation [124]. Arginase 1^high^ DCs could induce FoxP3^+^ Tregs and suppress T cell responses [125,126]. Arginase 1 conferred tolerogenic properties on DCs by promoting IDO expression via arginine catalytic degradation to ornithine [124]. Little is known about the role of arginase 2 expression and function in DCs. A previous report demonstrated that downregulation of arginase 2 by miRNA155 in murine bone marrow-derived DCs (BMDCs) could enhance T cell activation and proliferation [127]. In agreement with this result, a recent study revealed that human fetal cDC2s expressing arginase 2 mediated fetomaternal tolerance by promoting prenatal Treg induction [128]. It is therefore intriguing to further consider whether arginase 1 and arginase 2 have distinct or redundant function in DCs.

Glutamine is an abundant amino acid in plasma that plays a key role in regulating the immune system [129,130]. Glutamine is hydrolyzed to glutamate by glutaminase in mitochondria, and glutamate can be converted to glutamine by glutamine synthetase in the cytosol [130]. Anti-inflammatory effects of glutamine have been observed in severe inflammatory conditions including autoimmunity [131,132,133] Low glutamine levels in peripheral blood mononuclear cells (PBMCs) were associated with impaired mitochondrial respiration and higher SLE disease activity [134]. Metabotropic glutamate receptor-4 (mGluR4) is a glutamate receptor that is highly expressed on cDCs and pDCs [135]. Glutamate activation of mGluR4 signaling in DCs favored Treg induction, and consequently inhibited development of autoimmunity [135]. In addition, mGluR4 activation induced tolerogenic phenotypes in DCs through IDO-dependent signaling [136]. A metabolic connection exists between glutamine/glutamate and arginine [137], indicating that both of these metabolic pathways may contribute to tolerogenicity of DCs.

### 4.2. Glycolysis and Fatty Acid Metabolism

The tolerogenic status of DCs is also influenced by the metabolic circuits of glycolysis, FAO and OXPHOS, which are fueled by glucose, fatty acids, and glutamine, respectively. Human tolDCs generated by treatment with either vitD3 alone or dexamethasone and vitD3 displayed substantially increased catabolism (glycolysis, FAO and OXPHOS) in concert with increased mitochondrial respiration, glycolytic capacity and FAO activity [138]. Similarly, increased FAO and persistent OXPHOS resulted in a sustained tolerogenic state in murine BMDCs [139]. Inhibition of FAO interfered with the regulatory functions of tolDCs and partially restored their immunostimulatory function [138]. In the context of cancer, accumulation of cytosolic fatty acids impaired DC activation and promoted their immunosuppressive functions [140]. Therefore, lipid accumulation may support FAO and consequently skew DCs toward tolerogenic function.

### 4.3. Vitamin Metabolism

Metabolism of vitamin A and vitamin D has long been recognized to regulate DC function. Intestinal CD103^+^ DCs display elevated activity of ALDH, an enzyme catalyzing the conversion of retinol (vitamin A) to RA. CD103^+^ DCs capable of RA production promoted Treg generation in a TGF-β-dependent manner and suppressed Th17 polarization [32,141]. RA also converted DCs to tolerogenic phenotypes with high capacity to induce TGF-β-producing and IL-10-producing Tregs [142,143]. The inhibitory functions of RA-conditioned DCs may arise through enhanced arginase I activity [142]. Vitamin D3 has a strong effect on DC tolerogenicity, since vitD3-generated tolDCs could enhance Treg expansion and suppress autoimmune diseases in murine models [144,145]. Furthermore, vitD3 shifted activated moDCs from patients with autoimmune diseases toward tolDC phenotypes [70,144]. The mechanism underlying the vitD3-induced immunoregulatory functions of DCs requires the biological activity of IDO [145,146]. A recent study demonstrated that the glycolytic enzyme 6-phosphofructo-2-kinase/fructose-2,6-biphosphatase 4 (PFKFB1) played a specific role in vitD-induced metabolic reprogramming of tolDCs by regulating glucose metabolism [147].

Intracellular metabolism significantly regulates DC plasticity, because altered metabolic demands reprogram DCs from immunogenic to tolerogenic phenotypes (Figure 2). Thus, selective modulation of underlying cellular metabolism in DCs may provide new prospects for clinical therapy of SLE and other autoimmune diseases.

Tolerogenicity of DCs is controlled by the integration of glycolysis, FAO, OXPHOS, and amino acid catabolism together with vitamin A and vitamin D metabolism. The tricarboxylic acid cycle and IDO may represent central hubs of the metabolic processes in mitochondria and cytosol, respectively, governing the phenotypic and functional features of tolDCs. Abbreviations: 1,25VitD3, 1,25-dihydroxyvitamin D3; ALDH, aldehyde dehydrogenase; ARG, arginine; FAO, fatty acid oxidation; GLN, glutamine; GLU, glutamate; HO-1, heme oxygenase-1; IDO, indoleamine 2,3-dioxygenase; KNY, kynurenine; mGluR4, metabotropic glutamate receptor 4; ORN, ornithine; OXPHOS; oxidative phosphorylation; PFKFB, 6-phosphofructo-2-kinase/fructose-2,6-biphosphatase; RA, retinoic acid; TCA, tricarboxylic acid cycle; TRP tryptophan.

## 5. Clinical Implications of tolDCs for SLE

SLE is a multigenic autoimmune disease characterized by loss of T and B cell tolerance to self-antigens, dysregulated immune responses, production of autoantibodies and perturbed cytokine activities. Immune complex deposition and autoreactive T cell infiltration cause inflammation and tissue damage in multiple organs, and persistent disease activity over time is associated with significant morbidity and mortality. Similar to other autoimmune diseases, SLE therapies aim to restore immune tolerance to self-antigens. Clinical therapy of SLE is based mainly on anti-inflammatory and immunosuppressive drugs, which are non-specific and have broad systemic effects. Recently, several novel approaches for SLE therapy have been developed with improved specificity and efficacy and fewer side effects. TolDCs therefore have become promising targets and therapeutic tools because of their capacity to modulate immune responses in an antigen-specific manner. Cellular therapy with tolDCs and induction of tolDCs *in vivo* is of high interest for autoimmune therapy. Preclinical and clinical studies in SLE were summarized in Table 2 and Table 3.

Autologous tolDC-based therapies are currently under clinical trials for treatment of several autoimmune diseases including type I diabetes, rheumatoid arthritis, multiple sclerosis, and Crohn’s disease. However, little information is available regarding the use of tolDCs for SLE therapy, potentially due to the challenges of identifying SLE-specific antigens and the impaired functions of moDCs in SLE patients [10]. A few studies have explored the induction of tolerogenic phenotypes in autologous DCs from SLE patients and only tested their function *in vitro*. Combination treatment of MoDCs from SLE patients with vitD and dexamethasone could generate IL-10-producing tolDCs that induced potent Treg responses [12]. Another study demonstrated that MoDCs from SLE patients cultured in the presence of rosiglitazone and dexamethasone did not fully mature and were characterized by low proinflammatory cytokine production, indicating their tolerogenic phenotypes. In addition, these tolDCs were resistant to autologous apoptotic cell-induced maturation, and the apoptotic cell-loaded tolDCs could downregulate allogeneic CD4^+^ T cell responses [11]. Some probiotic bacteria have shown beneficial effects on inflammatory and autoimmune diseases. Culture MoDCs of SLE patients with probiotics, *Lactobacillus delbrueckii* and *L. rhamnosus*, could generate tolDCs with low maturation state. Furthermore, the expression of IDO and IL-10 was significantly unregulated while IL-12 was decreased in the probiotics-treated DCs [13]. These results help support the feasibility of generating tolDCs from peripheral blood monocytes of SLE patients. However, the stability of tolDC phenotype when infused back into the patients and the clinical outcome is lacking in SLE. 

Inducing tolDCs *in vivo* is more desirable therapeutic approach for autoimmune disease therapy. However, it is still difficult to selectively target DCs due to DC heterogeneity and lack of DC-specific molecules for drug delivery. Several targeted therapies for SLE have been developed, and some are relevant to tolDC induction. We thus survey some therapeutic approaches below from the point of view of *in vivo* tolDC generation and function.

SLE-specific antigens are required for Treg priming by DCs, and some examples of SLE-specific tolerogenic peptides which potently induce immune tolerance are available. Specific epitopes from nucleosomal histones induced immune tolerance in murine models of SLE. Therapy with the nucleosomal histone peptide H4_16–39_ delayed the onset of spontaneous lupus nephritis in SNF_1_ mice by suppression of autoreactive T cell and B cell responses [148]. In addition, immunization with nucleosomal histone peptide epitope, H4_71–94_, skewed splenic DCs and pDCs toward tolerogenic phenotypes, capable of substantial TGF-β production, which fostered Treg expansion and Th17 contraction in lupus-prone mice [149]. Human complementarity determining region 1 CDR1 (hCDR1) is a synthetic peptide based on the CDR1 sequence of a human anti-DNA monoclonal antibody with a 16/6 idiotype (16/6 Id) [173]. Treatment with hCDR1 successfully attenuated autoreactive T and B cell responses, and improved the clinical manifestations of murine and human SLE [150,151,152]. *In vivo* immunization with hCDR1 promoted the induction of tolerogenic DCs and FoxP3^+^ Tregs in lupus-prone mice [153,154]. Furthermore, hCDR1 (Edratide) demonstrated safety and efficacy in a phase II clinical trial in patients with active SLE, although it did not reach the primary outcome based on Systemic Lupus Erythematosus Disease Activity Index 2000 (SLEDAI-2K) and Adjusted Mean SLEDAI (AMS) [152]. The benefit of hCDR1 has been recently extended to clinical applications in primary Sjögren‘s syndrome, which is an autoimmune disease that mainly affects the exocrine glands. hCDR1 dampened expression of the proinflammatory cytokines, IL-1β and TNF-α, and upregulated the expression of the immunosuppressive molecules, IDO, TGF-β, and FoxP3, in PBMCs from primary Sjögren‘s syndrome patients [174].

Biological drugs targeting immune system-associated molecules involved in SLE pathogenesis have been developed. Their roles related to tolDCs are reviewed here. Adjunctive therapy with CTLA4-Ig (Abatacept and RG2077) has been evaluated in phase II clinical trials in SLE patients. CTLA4-Ig effectively inhibited T cell activation and proliferation based on its ability to outcompete CD28 for CD80/C86 ligation [71]. CTLA4-Ig binding to CD80/CD86 also transmitted a reverse signal to DCs, resulting in expression of IDO and induction of regulatory phenotypes in DCs [155,156]. In addition, combined treatment with IL-10-induced tolDCs and CTLA4-Ig efficiently ameliorated chronic lupus nephritis in MRL-Fas^lpr^ mice [157]. Nonetheless, a clinical trial study of CTLA4-Ig (Abatacept) and low dose cyclophosphamide combination therapy in patient with lupus nephritis has demonstrated a drug safety but was not effective to improve the clinical outcome of renal responses which may be due to the dose inadequacy of CTLA4-Ig [158]. Levels of CD40 ligand (CD40L or CD154) are significantly increased in the sera of SLE patients, and this increase is associated with disease severity [175,176]. Engagement of CD40L with CD40 on DCs contributes to DC activation, which consequently leads to amplification of T cell responses and B cell antibody production [177]. Anti-CD40L (dapirolizumab) therapy is currently under evaluation in a phase I clinical trial, and treatment with dapirolizumab can improve SLE severity [159]. In addition, treatment with anti–CD40 ligand antibody, BG9588, in patients with proliferative lupus nephritis reduced SLE pathogenesis (serologic activity and hematuria), suggesting the immunomodulatory action of the drug [160]. A recent study demonstrated that the CD40-CD40L interaction could abrogate Treg generation induced by intestinal CD103^+^ DCs in an autoimmune colitis mouse model [161]. Therefore, blockade of the CD40 pathway may facilitate the maintenance of tolerogenic CD103^+^ DC populations. pDCs are a major source of type I IFN, a key cytokine contributing to the pathogenesis of SLE. BCDA-2 is a specific marker of pDCs that plays a role in antigen internalization and presentation [178]. Targeting BCDA-2 on pDCs from SLE and cutaneous lupus erythematous patients, respectively, restored the tolerogenic state of pDCs by limiting type I IFN production which may enhance the clinical efficacy of SLE treatments [162,163,164]. In addition, activation of CXCR4 on pDCs by a small molecule, IT1t, stimulated immunoregulatory signaling in pDCs and prevented glomerulonephritis in a pristine-induced lupus-like model [165].

Some small molecule drugs developed to inhibit SLE pathogenesis and progression can also promote DC tolerogenicity by modulating the metabolic activity of DCs. The peroxisome proliferator-activated receptor gamma (PPARγ) agonists—rosiglitazone and pioglitazone—are being evaluated in ongoing phase II clinical trials for treatment of rheumatoid arthritis and SLE, respectively. Rosiglitazone-treated BMDCs exhibited decreased co-stimulatory molecule and MHC class II expression and had lower pro-inflammatory cytokine production, reflecting an immature tolerogenic phenotype. BMDCs treated with rosiglitazone were also capable of Treg induction, and adoptive transfer of these tolDCs ameliorated arthritis in a murine model of collagen-induced arthritis [166]. Furthermore, signal transduction via the Wnt5a-β-catenin-PPARγ axis increased FAO-mediated IDO activity in DCs in concert with increased Treg activation [167]. Adjunctive therapy with laquinimod, an aryl hydrocarbon receptor (AhR) agonist, is under evaluation in ongoing phase II clinical trials in SLE patients with active lupus nephritis. Aryl hydrocarbon receptors are ligand-activated transcription factors broadly expressed in mammalian cells including immune cells. Enhanced AhR expression and activity was observed in murine and human SLE, and activation of AhR was required for tolerance induction and attenuation of SLE [168]. Activation of AhR promoted IL-10, kynurenin and RA production in DCs resulting in Treg differentiation and expansion [169,170]. Similarly, activation of the kynurenine-AhR pathway could maintain IDO expression in tolDCs, suggesting a positive feedback loop involving IDO-AhR-IDO [179]. A selective AMP-activated protein kinase (AMPK) activator, metformin, is being evaluated in phase IV clinical trials for treatment of lupus flares in SLE patients [171]. VitD3-induced human tolDCs displayed upregulated AMPK activity and increased OXPHOS metabolism fueled by glucose, and human MoDCs treated with an AMPK activator (AICAR) exhibited a similar phenotype to that of VitD3-treated MoDCs [172]. Activation of AMPK also promoted FAO via PGC1α-PPARγ signaling [180].

## 6. Conclusions and Perspectives

Understanding the molecular and cellular basis of DC ontogeny and function provides fundamental insights into *in vitro* tolDC generation. The phenotypic and functional heterogeneity of DCs points to their great plasticity and diverse functions, which can be manipulated. A wide variety of protocols have been exploited for modulating tolDCs *ex vivo* and *in vivo*, which have resulted in heterogeneous properties of tolDCs [15,16,68]. Generation of tolDCs using a single tolerogenic agent also gave rise to tolDCs with different phenotypic and functional features [15,68]; these differences may result from variation in dosing and methods. Although there are limited data on specific strategies for tolDCs induction in SLE, the role of tolDCs in current therapies and clinical trials are under investigation. Scientists and clinicians alike will closely follow the progress of this innovative approach for SLE therapy. Applying different approaches either by *ex vivo* generation and adoptive transfer or *in vivo* induction of tolDCs in lupus mouse model gave quite promising results. However, clinical studies in SLE patients still have limited success. Most studies did not show clear benefit which is likely due to heterogeneity nature of SLE compared to other autoimmune diseases. The tolDC phenotypic and functional requirements for effective SLE therapy may depend on the mechanism of SLE pathogenesis and the stage of the disease progression. The genetic background of patients (e.g., polymorphisms of *FCGR2B*, *CTLA4*) and the condition of autologous monocytes derived from SLE patients may also impact the selection of tolDC-inducing protocols. It is necessary to further characterize biomarker that define responder group and conduct clinical trial based on subset of patients. So far, all of the clinical trials related to tolDCs in SLE use the *in vivo* induction which is more practical with less cost, however, the systemic and non-specific effect might be the main limitation of these approaches. It might be necessary to target those inducing substances directly to DC population to ensure the best efficacy and less toxicity e.g., nanoparticle-based treatment.

Another main issue in the clinical setting is the long-term maintenance of Treg in the patients to achieve complete and durable clinical responses. Almost none of the clinical studies in human investigated the presence of specific Treg in the patients after treatment. It is possible that lack of complete responses might come from the inability of tolDC to induce and maintain specific Treg in the patients. Genetic engineered tolDC or Treg approach is one alternative to solve this issue. The explosion interest in epigenetic regulation of immune cells have been reported and the combination of these modifications to maintain the stability of immune cell phenotypes might be another alternative to overcome current clinical dilemmas. Specific induction of tolerance via tolDCs will provide new prospects for more effective and less toxicity in a clinical therapy of SLE and other autoimmune diseases.

## Figures and Tables

**Figure 1 cells-08-01291-f001:**
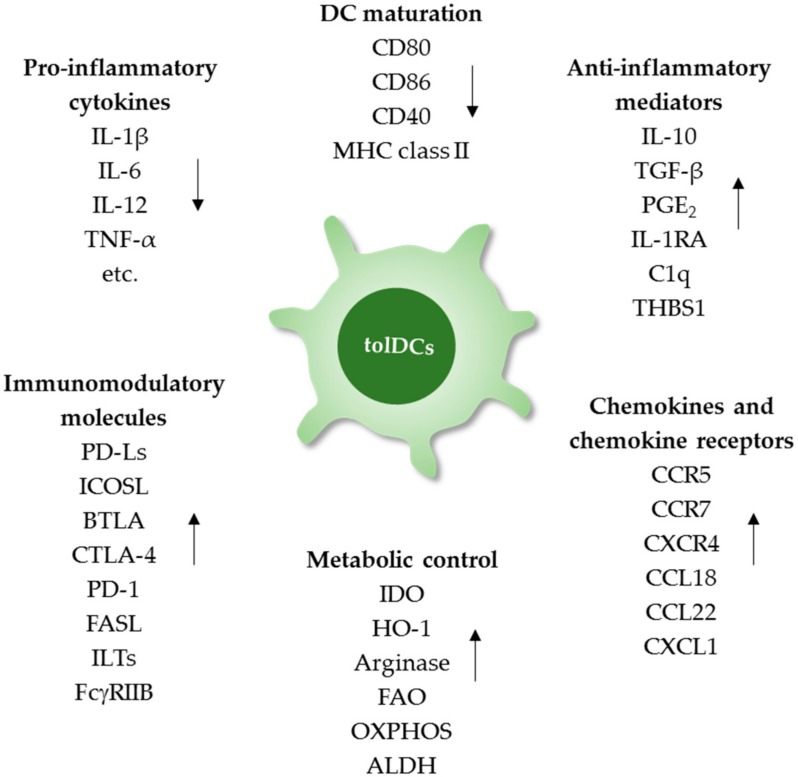
Desirable phenotypes of tolDCs.

**Figure 2 cells-08-01291-f002:**
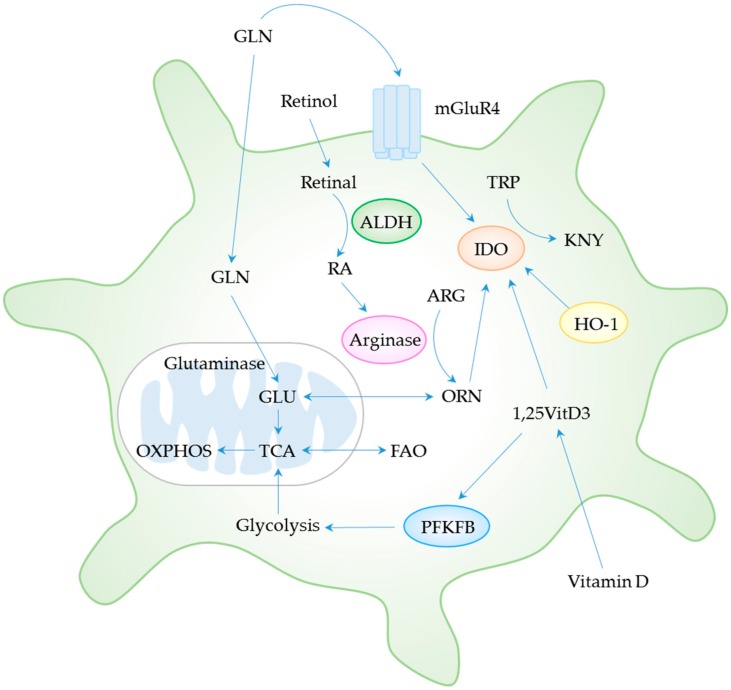
Metabolic circuits vital for DC tolerogenicity.

**Table 1 cells-08-01291-t001:** Mouse and human dendritic cell subsets and mechanisms involved in regulatory T cell induction.

Dendritic Cell Subsets	Surface Markers	Transcription Factors	Features of DC Subsets with Tolerogenic Functions	Mechanisms Involved in Regulatory T Cell Induction	Ref.
**Mouse**					
**1. Conventional DCs**					
**1.1 Lymphoid DCs**					
Resident cDC1	CD11c^hi^, CD8α^+^, CD11b^−^, Clec9a^+^, XCR1^+^, CD24^+^	Zbtb46, BATF3, ID2, IRF8	IDO^+^ cDC1 (induced via IFN-γ-inducible signaling)	IDO	[26,27]
			BTLA^+^CD205^+^ cDC1	BTLA and TGF-β	[28]
Resident cDC2	CD11c^hi^, CD8α^−^, CD11b^+^, DCIR2^+^, SIRPα^+^	Zbtb46, IRF4	No information	DCIR2	[29,30]
			IFN-γ-stimulated cDC2	Kynurenine (IDO-independent)	[31]
**1.2 Non-lymphoid DCs**					
Migratory cDC1	CD11c^+^, CD103^+^, CD11b^−^, Clec9a^+^, XCR1^+^, CD24^+^	Zbtb46, BATF3, ID2, IRF8	Mucosal cDC1	TGF-β, retinoic acid, IDO	[32,33]
Migratory cDC2	CD11c^+^, CD103^−^, CD11b^+^, CCR2^+^, SIRPα^+^, CD301b^+^, CX3CR1^+^	Zbtb46, IRF4	Retinoic acid-producing cDC2	ALDH, retinoic acid	[34]
			PD-L2^+^ cDC2	No information	[35]
**2. Non-conventional DCs**					
pDC	CD11c^lo^, B220^+^, Siglec-H^+^, BST2^+^	E2-2	IDO^+^ pDC	IDO	[36]
**Human**					
**1. Conventional DCs**					
Myeloid cDC1	CD141^+^, XCR1^+^, Clec9a^+^	Zbtb46, BATF3, ID2, IRF8	CD141^+^CD163^+^ DC-10	IL-10, ILT4/HLA-G, IDO	[37,38,39]
Myeloid cDC2	CD1c^+^, CD11b^+^, DCIR2^+^, SIRPα^+^, CX3CR1^+^	Zbtb46, IRF4	TLR4^+^ cDC2	IDO, IL-10	[40]
			Pulmonary cDC2	IL-10, IL-27, ICOSL	[41]
**2. Non-conventional DCs**					
pDC	CD45RA^+^, CD123^+^, CD303^+^, CD304^+^	E2-2	IDO^+^ pDC	IDO	[36]
			TLR9L-stimulated pDC	IDO	[42,43]

DC, dendritic cell; cDC, conventional DC; pDC, plasmacytoid DC, DCIR2, dendritic cell inhibitory receptor 2; SIRPα, signal regulatory protein α; BST2, BM stromal cell Ag-2; Zbtb46, zinc finger and BTB domain containing 46; BATF3, basic leucine zipper transcriptional factor ATF-like 3; ID2, inhibitor of DNA binding 2; IRF, interferon regulatory factor; E2-2, transcription factor 4; BLTA, B and T lymphocyte attenuator; PD-L2, programmed death ligand 2; DC-10, IL-10 producing DC; TGF-β, transforming growth factor-beta; IDO, indoleamine 2,3-dioxygenase; IFN-γ, interferon-gamma; ALDH, aldehyde dehydrogenase; IL-10, interleukin-10; ILT4, immunoglobulin-like transcript-4; HLA-G, human leukocyte antigen-G; ICOSL, inducible T cell costimulator ligand.

**Table 2 cells-08-01291-t002:** Implications of tolerogenic dendritic cells for SLE therapy.

Drug/Biologics	Phenotype(s) of tolDCs	Function(s) of tolDCs	Study Model	Ref.
**Ex vivo induction of tolDCs**				
Vitamin D + Dexamethazone	High IL-10	Treg induction	In vitro model in MoDCs from SLE patients	[12]
Rosiglitazone + Dexamethasone	Immature or semi-mature Resistant to autologous apoptotic cell-induced maturation	Inhibition of allogeneic CD4^+^ T cell response	In vitro model in MoDCs from SLE patients	[11]
L. delbrueckii L. rhamnosus	Semi-mature Increased IDO and IL-10	-	In vitro model in MoDCs from SLE patients	[13]
**In vivo** **induction of tolDCs**				
Tolerogenic nucleosomal peptide	Increased TGF-β	Treg expansion Contraction of Th17 Attenuation of autoreactive T cell and B cell	SNF_1_ mice	[148,149]
hCDR1	Decreased IL-1β and TNF-α Increased IDO and TGF-β	Treg expansion Attenuation of autoreactive T and B cells	BWF1 mice Clinical trial in active SLE patients	[150,151,152,153,154]
CTLA4-Ig	Increased IDO	Treg induction Attenuation of autoreactive T and B cells	MRL-Fas^lpr^ mice Clinical trial in SLE patients	[155,156,157,158]
Anti-CD40L	-	Treg induction	DC-LMP1/CD40 mice (autoimmune colitis) Clinical trial in SLE patients	[159,160,161]
Anti-BCDA-2	Decreased type I IFN in pDCs	-	In vitro model in pDCs from patients with CLE Clinical trial in SLE patients	[162,163,164]
CXCR4 agonist	Decreased type I IFN in pDCs	-	In vitro model in pDCs from SLE patients Pristane-induced lupus like model	[165]
PPARγ agonist	Immature or semi-matureIncreased FAO and IDO	Treg activation and induction	CIA mice Clinical trial in SLE patients	[166,167]
AhR agonist	Increased IL-10, kynurenins and retinoic acid Maintain IDO expression	Treg differentiation and expansion	In vitro model in MoM from patients with SLE and active SLE MRL-Fas^lpr^ mice Clinical trial in lupus arthritis and nephritis patients	[168,169,170]
AMPK activator	Increased OXPHOS and FAO	-	In vitro model in Human MoDCs Clinical trial in SLE patient as add-on treatment to treat lupus flare	[171,172]

DC, dendritic cell, tolDC, tolerogenic DC; MoDC, monocyte-derived DC; pDC, plasmacytoid DC; MoM, monocyte-derived macrophage; hCDR1, human complementarity determining region 1; CTLA4-Ig, cytotoxic T-lymphocyte–associated antigen 4-human Fc chimera; CD40L, CD40 ligand, BCDA-2, blood dendritic cell antigen-2; PPARγ, peroxisome proliferator-activated receptor gamma; AhR, aryl hydrocarbon receptor; AMPK, selective AMP-activated protein kinase; IL-10, interleukin-10; IL-1β, interleukin-1 beta; TNF-α, tumor necrosis factor-alpha; IDO, indoleamine 2,3-dioxygenase; TGF-β, transforming growth factor-beta; IFN, interferon; OXPHOS, oxidative phosphorylation; FAO, fatty acid oxidation; SLE, systemic lupus erythematosus; CLE, cutaneous lupus erythematosus; SNF^1^, lupus prone (SWR x NZB) F1 hybrid; BWF1, lupus-prone (NZBxNZW) F1 hybrid; DC-LMP1/CD40, transgenic DC constitutively express latent membrane protein 1/CD40; CIA, collagen-induced arthritis.

**Table 3 cells-08-01291-t003:** Clinical trials involving tolerogenic dendritic cells in SLE.

Drug/ Biologics	Sponsors/ Clinical Trial No.	Year	Study Population	Summary of Results	Challenges/ Future Development	Ref.
**Edratide (hCDR1)**	Teva Pharmaceutical industries/ NCT00203151	Sep 2005–Feb 2007	Phase II trial in Patients with active SLE (N = 340)	-The primary endpoints based solely on SLEDAI-2K and AMS were not met. -Improved BILAG, the Composite SLE Responder Index	-Need more clinical study to confirm result	The Prelude study
**Abatacept (CTLA4-Ig)**	NIAID/NCT00774852	Nov 2008–June 2014	Phase II add-on trial in active lupus nephritis (N = 134)	-No statistically significant differences in % complete response at 24 or 52 weeks	-Might need to test with higher dose	The Access Trial Group [158]
	Bristol-Myers Squibb/ NCT01714817	Jan 2013–May 2018	2-year long-term extension phase III trial in 405 patients with active class III or IV lupus nephritis	-No statistically significant differences in % complete renal response which is primary outcome -Earlier and more robust response	-Might be useful in different subset of SLE patients	The Allure trial
**BG9588 (Anti-CD40L Antibody)**	NIAMS/ NCT00001789	June 1999–May 2000	Phase I trial in lupus nephritis patients (N = 28)	-Terminated due to thromboembolic events -A short course of treatment showed some clinical improvement	-Additional studies will be needed to solve the issue of drug safety before evaluate its long-term effects	[160]
**BIIB059 (humanized mAb to BDCA2)**	Biogen/ NCT02106897	April 2014–May 2016	First in human trial Healthy control (N = 54), SLE with active skin lesions (N = 12)	-Favorable safety and PK profiles -Decreased expression of IFN response genes in blood -Reduced immune infiltrates in skin lesions	-Support the important role of pDC as target for skin manifestation	[164]
	Biogen/ NCT02847598	Oct 2016–Aug 2019	Phase II clinical trial in SLE patients with active skin manifestations and CLE patients (N = 264)	No result posted yet		
**Thiazolidinediones (TZD) (PPARγ agonist)**	NIAMS/ NCT02338999	Jan 015 Status: Active	SLE patients, Plan to enroll 88 participants	No result posted yet		
**Laquinimod (AhR agonist)**	Teva Pharmaceutical industries/ NCT01085084	July 2010–Dec 2012	Phase II clinical trial in SLE with active lupus arthritis (N = 82)	No result posted yet		
	Teva Pharmaceutical industries/ NCT01085097	July2010–Dec2012	Phase II clinical trial in SLE with active lupus nephritis (N = 47)	-62.5% of patients with active lupus nephritis received 0.5mg/day of laquinimod achieved renal response, compared to 33.3% of patients with placebo at 24 weeks	-Larger trial is needed to confirm result	https://www.tevapharm.com/
**Metformin (AMPK activator)**	Renji Hospital/ NCT02741960	May2016–Dec2018	Proof of concept trial of add-on metformin to conventional immunosuppressants in lupus flares in SLE patients (N = 113)	-Metformin as an add-on can reduce clinical flares, prednisone exposure, and body weight	-Promising choice, larger trial is needed to confirm result	[171]

hCDR1, human complementarity determining region 1; SLEDAI-2K, Systemic Lupus Erythematosus Disease Activity Index 2000; AMS, Adjusted Mean Systemic Lupus Erythematosus Disease Activity Index-2K; BILAG, British Isles Lupus Assessment Group; CTLA4-Ig, cytotoxic T-lymphocyte–associated antigen 4-human Fc chimera; NIAID, National Institute of Allergy and Infectious Diseases; CD40L, CD40 ligand; NIAMS, National Institute of Arthritis and Musculoskeletal and Skin Diseases; BCDA-2, blood dendritic cell antigen-2; PK, Pharmacokinetics; IFN, interferon; CLE, cutaneous lupus erythematosus; PPARγ, peroxisome proliferator-activated receptor gamma; AhR, aryl hydrocarbon receptor; AMPK, selective AMP-activated protein kinase.

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
