# Peer review of "Current Paradigms of Tolerogenic Dendritic Cells and Clinical Implications for Systemic Lupus Erythematosus"

_cells, 2019, doi:10.3390/cells8101291_

Round 1

Reviewer 1 Report

This work by Ritprajak and colleagues provides a broad overview on tolerogenic dendritic cells and possibilities to use them in clinical, autoimmune settings. Although the information is extensive and detailed several critical points remain unclear and need to be addressed:

Although chapters 2 contain massive amount of information related to subsets and phenotypical features of several DC populations, what their ontogenetic relationships are have been poorly mentioned. Overall, the reader do not get a clear understanding as to whether: a, immunogenic and tolerogenic DC arise from same or from different precursors; b, whether their phenotype is fixed or flexible, e.g. whether an immunogenic phenotype can give raise to a tolerogenic one, and viceversa; c, if this is the case, what environmental factors may contribute to maintain or to modify their features? All these aspects might be relevant when inducing the required phenotype in vitro or when tracking the fate of tolDCs in vivo and their ability to reduce autoimmunity.

The anatomical location of different DC subsets in relation to their function has barely been mentioned. Do tolDC acquire their tolerogenic properties due to a distinct anatomical location?

Is there any correlation in between surface marker expression and modulation and tolerogenic functions?

By far Table 2 represents the most important piece of information in this article as it summarizes current efforts to move forward with tolDC from bench to bedside. For this reason, the information needs a deep review as it contains multiple, serious mistakes: work cited in line 1 refers to reference 136 and not 135; work in line 2 refers to reference 137 and not 136; unclear whether or not reference 135 should be included in this table; reference 138 refers to the use of probiotics and not to tolerogenci nucleosomal peptides; reference 152 refers to an immunohistology study of CD40 and CD40L expression on tissues. Therefore, it is unclear why the authors mention anti-CD40L as “drug” and Treg as “Function of tolDCs”; reference 159 on table never mentions the word “Treg”. All this (may be there are more) unnerved the patience of this reviewer.

Additional comments: Chapter 2 appears in text two times as such, in page 2 (DC subsets) and page 4 (Phenotype)

Reviewer 2 Report

Several comprehensive reviews and monographs have appeared in the recent years on new therapies for SLE, aimed at avoiding the long-term use of immunosuppressive drugs, which display unwanted secondary effects. The present review article is specially focused on possible application of tolerogenic dendritic cells.

Quite regrettably, the presentation of text is sometimes superficial, with typical sentences that can be read in many other reviews and articles (especially in the first introductive paragraphs) without any critical thinking. The tone that is set remains essentially descriptive and the authors express no opinion or more personal ideas. They do not mention any limitations or specific criticisms in the analyses. These aspects are important in a review article because they can orientate future research.

This article contains one small figure only. The two tables contain “classical” information. Illustrations are important in such comprehensive reviews as they support and help elaborating ideas and propose new mechanisms.    

A table listing the clinical trials that have been done and /or are currently performed (NIH number, number of patients, application, limitation, hopes and other informative facts) would be useful and would give some originality and more impact in this article (see the ambitious title).

A table listing the small molecule drugs and peptides designed to inhibit lupus pathogenesis and progression and promote DC tolerogenicity would fix the idea and clarify the text in page 10. Here again at the end of the paragraph (page 11, lines 401), personal views of the Authors regarding this strategy would be useful to distinguish this articles from several others that have been published on the same subject.

There are no avenues that are proposed in the section name “Perspectives” (page 11).

Minor comments

-there are two paragraphs numbered “2” (pages 2 and 4)

-separate the legends from the text. In the present form, they are intricate, which is distracting for the reader.

-Table 2: patients are not “study model”.

-Table 2: it is uneasy to understand why on the same line there are the name of mouse models and patients. Separate the two?

Reviewer 3 Report

In this manu, Patcharee and other authors discuss the features of tolerogenic dendritic cells and their potential applications in SLE clinics. The manu is clearly divided into 6 major parts, focusing on the detailed introductions of the phenotypes, functions, metabolites of tol-DCs. Here I list some points that the authors may take to improve the quality of the manu and to make it more readable. 

The manu is focusing on tol-DCs. However, after reading through the paper, I still don’t know exactly what is tol-DCs… There is no clear definition or introduction of it, though a lot details have been discussed. I know tol-DCs are not a specific cell lineage but mostly induced subsets of DCs, which is difficult to clarify or define. But you have to make it as clear as you can to those readers who have never known what’s tol-DCs from the very beginning. Another important thing I need to bring out is the references. Many sentences everywhere are without references. This is a review paper and most of the words you say need evidences to support. Please add references. The general structure of this manu is fine. The major parts are clear to tell. The only suggestion is to combine part 2 ‘DC subsets in immune tolerance’ with part 3 ‘Phenotypic and functional signatures of tol-DCs’. The table 1 is good in illustrating the details, so you can simplify the words in describing them. Please re The abstract, as a whole, is not attractive in either concluding the major contents in this review or bringing out any interesting concepts. Please rewrite it. In particular the last sentence, I cannot understand it. For the Introduction part (line 28-45), there are no any references. Almost every sentence in this part needs citations. Line 32-33: ‘DCs … induced by inflammatory signals such as pathogens …’. I am not sure if it’s accurate to say this way. Maybe ‘… induced by pathogens and inflammatory signaling molecules’ is better. The second paragraph of the introduction (line 38-45) is not written in a good logic. Actually, I don’t agree with some points in this paragraph, so please add evidences to support your every sentence. After reading the manu, another strong feeling is that the authors are packing tons of data but they don’t know what are the key points. In other words, the authors need to think more and have some opinions on what are done and what’s going on in this research field. In the part of ‘Metabolic control of DC tolerogenicity’, it will be more readable if the authors add some sub-titles.

Round 2

Reviewer 1 Report

.